# Aflatoxin Contamination, Exposure among Rural Smallholder Farming Tanzanian Mothers and Associations with Growth among Their Children

**DOI:** 10.3390/toxins15040257

**Published:** 2023-04-01

**Authors:** Wanjiku N. Gichohi-Wainaina, Martin Kimanya, Yasinta C. Muzanila, Nelson C. Kumwenda, Harry Msere, Mariam Rashidi, Omari Mponda, Patrick Okori

**Affiliations:** 1International Crops Research Institute for the Semi-Arid Tropics (ICRISAT), Chitedze Agricultural Research Station, Lilongwe P.O. Box 1096, Malawi; 2Department of Nutritional Sciences, College of Human Sciences, Texas Tech University, P.O. Box 41270, Lubbock, TX 79409-1270, USA; 3School of Life Science and Bio-Engineering, The Nelson Mandela African Institution of Science and Technology, Arusha P.O. Box 447, Tanzania; 4Department of Biosciences, Sokoine University of Agriculture (SUA), Morogoro P.O. Box 3038, Tanzania; 5Tanzania Agricultural Research Institute (TARI, Naliendele), Mtwara P.O. Box 509, Tanzania

**Keywords:** aflatoxins, Tanzania, cereals, nuts, food safety and security

## Abstract

Recently, aflatoxin exposure especially through maize and groundnuts has been associated with growth impairment in children. Infants and children are considered to be more susceptible to toxins because of their lower body weight, higher metabolic rate, and lower ability to detoxify. On the other hand, for women of reproductive age, aflatoxin exposure may not only affect their health but also that of their foetus in the case of pregnancy. This study focused on investigating AFB_1_ contamination in maize and groundnut from respondent households, exposure among women of reproductive age and associations of aflatoxin contamination with growth retardation among children in Mtwara region, Tanzania. The highest maximum AFB_1_ contamination levels from all samples obtained were in maize grain (2351.5 μg/kg). From a total of 217 maize samples collected, aflatoxins were above European Union (EU) and East African Community (EAC) tolerable limits in 76.0% and 64.5% of all samples. Specifically, maize grain had the highest proportion of samples contaminated above tolerable limits (80.3% and 71.1% for EU and EAC limits). Groundnut had 54.0% and 37.9% of samples above EU and EAC maximum tolerable limits. The lowest proportion of contaminated samples on the other hand was for bambara nut (37.5% and 29.2% for EU and EAC limits, respectively). Aflatoxin exposure in our surveyed population was much higher than previous observations made in Tanzania and also higher than those observed in Western countries such as Australia and the USA. Among children, AFB_1_ concentration was associated with lower weight for height z scores and weight for age z scores in the univariate model (*p* < 0.05). In summary, these results indicate the seriousness of aflatoxin contamination in foods commonly consumed in the vulnerable population assessed. Strategies both from the health, trade, and nutrition sectors should therefore be designed and implemented to address aflatoxin and mycotoxin contamination in diets.

## 1. Introduction

One-third (34%) of Tanzanian children under age 5 are stunted (short for their age), with Mtwara having a prevalence higher (38%) than the national average [1]. Recently, aflatoxin exposure, especially through maize and groundnuts, has been associated with growth impairment in children [2,3,4]. Aflatoxins are immune suppressive and are also thought to be one of the major causes of liver cancer and can affect other organs as well [5]. There are four different forms of aflatoxins, such as aflatoxin B_1_ (AFB_1_), B_2_ (AFB_2_), G_1_ (AFG_1_), and G_2_ (AFG_2_), and all of them are known to have effects on health, with AFB_1_ being the most potent form [6].

Infants, young children, and women of reproductive age are especially important to consider in terms of aflatoxin exposure. This is because infants and children are considered to be more susceptible to different toxins than adults, because of their lower body weight, higher metabolic rate, and lower ability to detoxify [7]. On the other hand, for women of reproductive age, aflatoxin exposure may not only affect their health but also that of their foetus in the case of pregnancy. In fact, high concentrations of AF in cord blood (i.e., in utero exposure) have been associated with subsequent infant linear growth faltering in Africa [8,9]. Due to the high prevalence of mycotoxin contamination in cereals and protein crops, the mother and the developing foetus can be at special risk. Besides their growth retardation, carcinogenicity and immunosuppression effects [10], aflatoxin also has negative economic impacts [11]. This is because there are various regulatory limits for aflatoxin-prone grain intended for human consumption. These regulations may be national, regional, or international depending on the country [12]. For example, the current maximum tolerable limits for total aflatoxins in maize and groundnut in the EU, EA, and the USA are 4, 10, and 20 µg/kg, respectively [13,14,15]. For farmers that grow these grains for sale in country or export, failure to meet these regulations has dire impacts on income.

To determine aflatoxin exposure, both the aflatoxin concentration in a sample and the consumption pattern of a food are considered. For this reason, maize and groundnut are important aflatoxin-prone crops to consider due to their high daily consumption [16].

In Tanzania, maize is of particular concern as the dependence on maize as a main staple food is high. Currently the per capita consumption is approximated at 429.4 g/person/day, which is less than the Tanzanian Food and Nutrition Center (TFNC) recommended per capita consumption of 771 g/person/day for non-dehulled maize flour or 790 g/day for dehulled maize flour [1]. When groundnut is considered, the approximated per capita consumption is 15 g/person/day [1]. Using these consumption rates, dietary exposure can then be estimated to obtain parameters such as probable daily intake (PDI), average probable daily intake (APDI) and maximum probable daily intake (MPDI), which are usually expressed in ng/kg·bw/day [17]. 

This study focused on investigating AFB_1_ contamination in maize and groundnut from respondent households, exposure among women of reproductive age and associations of aflatoxin contamination with growth retardation among children in Mtwara region, Tanzania. Within this assessment, we investigated the percentage of collected samples of maize, sorghum, and various nuts contaminated beyond tolerable limits that is likely to lead to food and nutrition insecurity and monetary losses if the present-day contamination benchmarks were enacted in both local and international markets.

## 2. Results

### 2.1. Socio Demographic Characteristics

Of the surveyed households, 20.1%. of the children were stunted and 40.4% of caregivers were overweight or obese. This stunting figure is lower than the national average of 34% and the Mtwara average of 38%. The overweight/obesity figures we observed are higher than the national average of 28% [18]. In terms of diet quality, 4.5% of children and 32.2% of women met the minimum dietary diversity scores. The last demographic health survey observed that 12.9% of children aged 6–23 months met minimum dietary diversity standards [18]. Over 80% and 7.4% of surveyed households were reported as using maize and groundnut as part of their complementary food, respectively. Only 13.9% of children aged 6–23 months were reported to have consumed vitamin-A-rich fruits and vegetables in the last 24 h and 32.1% as having consumed other fruits and vegetables in the 24 h preceding the survey (Figure 1). Similarly, less than half of the women (42.6%) of reproductive age consumed dark-green leafy vegetables, 1.3% other vegetables, 24.7% other fruits and 2.1% other vitamin-A-rich fruits and vegetables (Figure 2).

In terms of household characteristics important for aflatoxin management and messaging to smallholder farmers, less than 50% owned grain stores and the radio was the most common communication asset owned (51%).

Other socio-demographic characteristics of surveyed households are presented in Table 1.

### 2.2. Aflatoxin Levels

The highest maximum AFB_1_ contamination levels from all samples obtained were in the maize grain (2351.5 μg/kg). Mean aflatoxin levels in maize grain were 156.5 μg/kg (Table 2). When crop AFB_1_ levels were compared to those of groundnut, maize grain had significantly higher AFB_1_ levels (*p* = 0.003). We then evaluated the proportion of samples with AFB_1_ levels exceeding maximum tolerable limits in the EU and EAC of 2 μg/kg and 5 μg/kg, respectively. From a total of 217 maize samples collected, aflatoxins were above EU and EAC tolerable limits in 76.0% and 64.5% of all samples. Specifically, maize grain had the highest proportion of samples contaminated above tolerable limits (80.3% and 71.1% for EU and EAC limits). Groundnut had 54.0% and 37.9% of samples above EU and EAC maximum tolerable limits. Lowest proportion of contaminated samples on the other hand was bambara nut (37.5% and 29.2% for EU and EAC limits).

### 2.3. Aflatoxin Exposure

The probable daily intake of aflatoxins, assuming that all maize in a household was for human consumption, ranged between 0.43 and 14,571 ng/kg·bw/day when an average maize per capita consumption per day of 429 g/day/person was considered according to method described by Abt Associates (2012). When the maize-consumption rate recommended by the Tanzanian Food and Nutrition Center (TFNC) of 771 g/person/day was considered, exposure rates were considerably higher, ranging between 0 and 26,162 ng/kg·bw/day (Table 3). The average probable daily intake was 1133 ng/kg·bw/day using the Abt average per capita consumption per day of 429.4 g/person/day; and 2034 ng/kg·bw/day using the TFNC-recommended consumption rate of 771 g/person/day (Table 3). Finally, the maximum probable daily intake was 3906 and 7699 ng/kg·bw/day for the 90th percentile using the Abt- and TFNC-recommended consumption rate of maize, respectively (Table 3). 

Groundnut consumptions considering the Abt Associates (2012) per capita consumption per day for Mtwara was 15.1 g per day. Exposure to AFB_1_ ranged from 0 to 375 ng/kg·bw/day, with an average of 14.1 ng/kg·bw/day, and a 90th percentile of 18 ng/kg·bw/day (Table 3).

### 2.4. Association of Aflatoxin Levels with Growth Markers

In assessing risk of stunting and underweight, variables of exposure included in models were: age in months, sex, diet diversity score, wealth quintile and AFB_1_ contamination. The concentrations of AFB_1_ were not associated with the odds of stunting in both the multivariate and univariate model. On the other hand, a unit increase in contamination was associated with an increase in height for age z score in the multivariate model (*p* < 0.05; Table 4), though the increase was not of public health significance (<0.1 z score). Moreover, AFB_1_ concentration was associated with lower weight for height z scores and weight for age z scores in the univariate model (*p* < 0.05; Table 5 and Table 6). In both the univariate and multivariate model, AFB_1_ concentrations were not associated with the odds of wasting or underweight.

## 3. Discussion

Our study aimed to determine the occurrence of aflatoxins in commonly consumed cereals and nuts from Mtwara region in Tanzania. Within the analyses we also determined the proportion of samples contaminated beyond the EAC and EU maximum allowable limits. In addition, we conducted an exposure assessment in mothers and finally a risk assessment for developing stunting, wasting, and underweight among their children less than 24 months. The highest maximum AFB_1_ contamination levels from all samples obtained were in maize grain (2351.5 μg/kg). From a total of 207 maize samples collected, aflatoxins in 69.2% and 48.9% of all collected samples were above EU and EAC maximum tolerable limits. Groundnut had 42.5% and 20.7% of samples above EU and EAC maximum tolerable limits. The probable daily intake of aflatoxins in maize ranged between 0.43 and 14,570.5 ng/kg·bw/day when using an average maize consumption rate of 429.4 g/day/person. Using the maize consumption rate of 771 g/person/day, exposure rates were much higher. When groundnut consumption was considered, exposure to AFB_1_ was lower than that from maize consumption. Finally, AFB_1_ at household level was associated with increased risk of stunting and lower weight for height z score and weight for age z score (*p* < 0.05). 

In order to understand the impact of aflatoxin exposure on a population, dietary habits, especially consumption of crops at risk of aflatoxin contamination, needs to be assessed. In this population we observe a common use of maize in complementary foods with a low proportion of children reported to meet the minimum acceptable diet. Poor dietary diversity is also observed among women of reproductive age. The Tanzanian population relies on maize as a staple food for both adult diets and in complementary foods [19,20]. Harvesting of maize is ideally performed when kernels are dried to below 14% moisture content to facilitate incorporation into household recipes [21]. However, maize and maize products are highly susceptible to aflatoxin contamination and therefore populations that consume maize frequently are highly exposed to aflatoxins and their impacts. Differences in aflatoxin contamination between different forms of maize has previously been observed [22]. On the whole, high levels of maize contamination have been observed in several studies. For example, it has been estimated that Kenyans are exposed to aflatoxins in the range of 4.3–554 ng·kg^−1^·bw·day^−1^ [22] and Tanzanians in the range of 3.0–1092.6 ng·kg^−1^·bw·day^−1^ [19]. Exposure in our surveyed population was much higher than these previous observations and also higher than those observed in Western countries such as Australia and USA of 0.8 ng and 0.26 ng·kg^−1^·bw·day^−1^, respectively, from aflatoxin-contaminated maize and maize-based products [23]. Considering that this does not take into account additional aflatoxin exposure from groundnuts, the Tanzanian population we investigated herein are highly susceptible to developing adverse health outcomes in both childhood and adulthood if early measures to mitigate aflatoxin exposure are not practiced or enforced. 

Indeed, daily consumption of aflatoxin-contaminated food and food products for a considerable length of time may culminate in the development of hepatocellular carcinoma (HCC) [24]. The current burden of HCC attributable to aflatoxin exposure in developing countries is not widely estimated; thus, there is need for increased exposure and risk assessment studies in these regions. Two previous studies have investigated exposure assessment and population risk for primary liver cancer (cancers/year per 100,000 population) in Tanzania. One study investigated exposure based on beer consumption and observed a risk of 33.1 cancers/year per 100,000 population among adults and the other 2.95 cancers/year per 100,000 population among children based on the consumption of maize. In our study, we did not investigate the risk of liver cancer, since we did not have data on the actual consumption of maize and groundnut. Our exposure estimates however do indicate a need to investigate this association to determine the exact contribution of aflatoxin exposure to cancer incidence.

Evidence has recently emerged that exposure of children to aflatoxin-contaminated foods is associated with growth faltering and stunting as a result of chronic exposure [25]. In our study, AFB_1_ contamination in the household was not associated with stunting, wasting, or underweight but was associated with an increased height for age z score and a decreased weight for height z score, although unit changes were not of public health significance. It is likely that we did not observe any strong associations due to our low sample size and a lack of individual exposure data as we did not conduct a detailed dietary consumption assessment. In assessing associations with child growth, growth velocity assessed via a longitudinal study would have been a more informative measure of the association between mycotoxin contamination and growth failure. Understanding the biology underlying the relation between aflatoxin and growth is also crucial to investigate. This is because it is hypothesized that aflatoxins can alter mucosal barriers and affect resistance to intestinal infections [26] and micronutrient absorption [26,27] leading to growth impairment. Besides this, investigating the presence of multiple mycotoxins is imperative as mycotoxins are common in grains as mixtures, particularly with fumonisins in maize [28]. The role of possible interactions between these co-contaminants in the underlying mechanisms of growth impairment is of further interest especially in populations where child growth is of public health importance, such as in Tanzania. Another important investigation is one of dietary intake as highly contaminated food is likely a marker for food of poor nutritional quality and thus a reduced dietary intake of nutrients would be the actual underlying cause of the association between mycotoxin contamination and impaired growth. In this study we did not assess blood micronutrient status and could therefore could not ascertain this association. We did however assess diet quality via dietary diversity assessment and observed poor diet quality with a high proportion of children having maize as part of their complementary food and the mother–child dyads assessed as having a low intake of fruits and vegetables. Although groundnut was not as commonly consumed, they still remain of importance due to their high contamination. To fully distinguish the effects of aflatoxins or mycotoxins from other confounding factors in the diet, it would be best to conduct a randomized intervention study where the impact of lowering aflatoxin exposure on immunity, growth, and disease susceptibility can be assessed. This would also allow a better understanding of the relative contribution of aflatoxin to growth impairment in relation to other determinants of nutrition status and health such as sanitation and hygiene. 

Besides the health impacts of aflatoxins, they are a major constraint on accessing export markets. Potential high-income markets for groundnuts have strict food safety regulations. Such regulations are also coming into place for maize contamination within the EAC [29]. A high contamination above allowable limits may lead to a decline of exports and therefore a loss of income for farmers. Economic yield losses may be up to 100% if the aflatoxin levels exceed stipulated levels [30]. Where grain meant for export fails to meet set standards, aflatoxin-laden grain remains within the local food system further affecting nutrition and health of populations already consuming poor diets. This is because regulations are likely to have a limited effect on the foods consumed on the farm or sold in informal markets. Along with regulatory and other post-harvest measures, there is an urgent need to raise awareness and educate parents/caregivers on the aflatoxin health risks associated with complementary foods and the appropriate strategies to minimize contamination. Several studies on knowledge of aflatoxin contamination among households in Central Tanzania have been carried out, showing a low proportion of respondents possessing knowledge of aflatoxin contamination and its effects [31,32,33,34]. This lack of knowledge may contribute to high aflatoxin contamination observed in corresponding respondent households. Creating pervasive awareness from production to consumption about mitigating aflatoxin exposure and its effects, explicitly in at-risk communities, is vital for management. Well-designed information dissemination campaigns using appropriate channels would serve as the foundation for initiating and sustaining behaviour changes that mitigate aflatoxin contamination. The use of technology to ensure widespread information dissemination would thus be considered especially in communities that have appreciable literacy. We observe that 70% of caregivers in this survey report having mobile phones—a tool that may be useful in accessing aflatoxin mitigation messages.

Among the mitigation approaches that could be promoted is proper storage. Indeed it has been observed that poor storage decreases maize stocks by 150–250 kg per every ton stored [35]. Previous studies in Central Tanzania have observed that granaries in the storage area, mostly made of mud and plant materials, were in poor condition and therefore unable to eliminate insect pests in storage [35]. Addressing harvesting and handling practices is critical for protecting grain against contamination, though promoted practices need to be affordable and accessible to smallholder farmers.

This work assessed aflatoxin contamination in two important food crops in Tanzania. Although the data presented on stunting risk are based on a number of assumptions, these findings are important in stimulating more research, especially within the African continent, in order to develop more accurate exposure and risk assessments. The data, however, do indicate the seriousness of aflatoxin contamination in foods commonly consumed in the vulnerable population assessed. Strategies both from the health, trade, and nutrition sectors should therefore be designed and implemented to address aflatoxin and indeed mycotoxin contamination in diets. In the long term, this also indicates current and potential income losses where the cultivation of some of these crops is intended for export markets, thus highlighting the need to support continued efforts to reduce aflatoxin contamination.

## 4. Method

### 4.1. Study Design, Ethical Approvals, and Consent

The study in the Mtwara region was a cross-sectional study designed to assess aflatoxin contamination at the household level. All households selected had children less than 24 months old. Ethical approval for this study was obtained from the National Institute for Medical Research, approval number NIMR/HQ/R.8a/Vol.IX/2998. Eligible participants provided informed written consent. Additional approvals required from local authorities were obtained from administrative officials and medical officers. 

### 4.2. Study Sites

Households (*n* = 250) in the Mtwara region of Tanzania were selected purposively from Masasi and Nanyumbu districts (Figure 3). The villages and households selected were part of the McKnight project zone of influence aimed at improving groundnut productivity for food and nutrition security. Mtwara is 38 m above sea level. In the Mtwara region, the summers are much rainier than the winters. The average temperature in Mtwara is 25.9 °C and precipitation is about 970 mm per year [36]. Mtwara experiences one rainy season, with early rains in December and late rains in April and main rains from January to March. In the Mtwara region, approximately 90% of the economically active population is involved in agriculture, primarily cashew nut production [37]. In fact, of the total harvested area of cashews, Mtwara comprised 53.8% [38]. The latest available data from the Tanzania Demographic Health Survey (TDHS) indicates that 47.1% of women in Mtwara suffer from anaemia which was higher than the national average of 25% [39]. In terms of children’s nutrition and health, the latest Tanzania Demographic Health Survey and Malaria Indicators Survey (TDHS-MIS) indicates that the prevalence is stunting and of public health significance at 22.3% [40]. In addition, 58.6% of children 6–59 months were previously observed to be anaemic in the previous national survey [18]. These maternal and child indicators may be due to poor maternal and child diet quality.

### 4.3. Dietary Assessment

Dietary diversity was evaluated and then the minimum dietary diversity for women (MDD-W) calculated. This is an indicator that is calculated to determine whether women of reproductive age (15–49 years) have consumed five out of ten defined food groups at a minimum in the past 24 h. In detail, respondents from 15 to 49 years of age were selected for the study. An open 24 h recall was then used to obtain a record of foods and beverages consumed in the past 24 h. For mixed dishes, the main ingredients utilized in preparation were probed from respondents to identify the food groups consumed. The 10 MDD-W score was then obtained based on the reported consumption of the following food groups: (1) grains, white roots and tubers, and plantains; (2) pulses (beans, peas and lentils); (3) nuts and seeds; (4) dairy; (5) meat, poultry and fish; (6) eggs; (7) dark-green leafy vegetables; (8) other vitamin-A-rich fruits and vegetables; (9) other vegetables; and (10) other fruits. The MDD-W score therefore ranged from 0 to 10. To determine whether MDDW-was met, each woman was then coded “yes” or “no” using a cut-off of five to determine whether they met the recommended MDD-W [41].

The dietary assessment questionnaire for children was administered to the caregiver. The minimum dietary diversity (MDD) score for children 6–23 months old is also based on a 24 h open recall with foods classified into eight food groups: (1) breastmilk; (2) cereals, roots and tubers; (3) legumes and nuts; (4) milk and its derivatives; (5) meat products (meat, poultry, offal, and fish); (6) eggs; (7) vitamin-A-rich fruits and vegetables (leafy green vegetables, yellow fruits and vegetables); and (8) other fruits and vegetables [42]. The dietary diversity score (DDS) was defined as the total number of food groups consumed by the child in the past 24 h. Based on the WHO guidance, a child with a DDS < 5 was classified as having low dietary diversity; otherwise, they were considered to have adequate dietary diversity [43]. 

### 4.4. Household Questionnaire and Anthropometric Measurement

A questionnaire was prepared and administered to assess various household characteristics as well as socio-demographic characteristics, such as the birth date and sex of the child. Household characteristics included were, for example, household size and asset ownership. All personnel recruited to administer the questionnaire either had a bachelor’s degree or had prior experience administering questionnaires in this environment. Prior to the implementation of the study, they were coached and their skills in questionnaire administration examined based on a pretesting activity. All interviews took place in Swahili. After each survey day, submitted questionnaires were assessed for completeness both by the study supervisors and the principal investigator, prior to data analyses. 

Anthropometric measures of mother–child dyads were then recorded. Each child’s height was measured in reclining position using the recommended height board designed by UNICEF. All height measurements were recorded to the nearest 0.1 cm. On the other hand, the height of caregivers was measured in the standing position without shoes to the nearest 0.1 cm, using a portable stadiometer. The weights of both caregivers and children were measured in duplicate to the nearest 0.1 kg using an electronic scale (SECA Model 803, Hanover, MD, USA) [44].To calculate height-for-age z scores (HAZ), weight-for-age z scores (WAZ), and weight-for-height z scores (WHZ), WHO Anthro software version 3.2.2 [45] was used. Based on WHO criteria, a z score of less than –2 for HAZ indicates stunting; with the same cut-off for WAZ indicating undernutrition and for WHZ indicating wasting. Furthermore, BMI was classified as underweight (<18.5 kg/m^2^), normal weight (18.5 kg/m^2^ to <25 kg/m^2^) or overweight and obese (≥25 kg/m^2^) based upon pre-defined criteria [46]. 

### 4.5. Sample Collection

In addition to the described training on questionnaire administration, study personnel were also taught how to collect cereal and nut samples. This was undertaken through practicing with a collection of samples as part of the pretest as part of their competence assessment. Both flour and grain samples were collected from what was available in the household food-preparation area. If foods were not available or had run out in this part of the house, a sample from the storage bag or container the household would consume next was collected. During this collection, samples were gathered from different sections of the farmer’s storage vessel and thoroughly mixed. Where maize cobs were sampled from different parts of a storage vessel, they were shelled and the grains thoroughly mixed. Samples (1000 g) were then analysed for levels of aflatoxin as described below.

### 4.6. Determination of Aflatoxin Concentrations

From every 1000 g of collected samples, a subsample of 200 g was collected and ground into a fine powder using a heavy-duty grinding machine (Robot Coupe, South Perkins, Ridgeland, Mississipi). Maize grain samples were milled using a Cyclone mill (UDY-3010-014). From the ground samples, two equal portions were obtained, then one portion was triturated in 70% methanol (*v*/*v* 70 mL absolute methanol in 30 mL distilled water) containing 0.5% *w/v* potassium chloride in a blender, until thoroughly mixed. 

The mixture obtained was transferred to a conical flask and shaken for 30 min at 300 revolutions per minute (rpm). The extract was then filtered using Whatman No.41 filter paper (Sigma-Aldrich, St. Louis, MO, USA) and then diluted 1:10 in phosphate-buffered saline containing 500 µL/l Tween20 (PBS–Tween, Sigma-Aldrich, St. Louis, MO, USA). Finally, the filtrate was analysed for AFB_1_ using an in-house indirect competitive enzyme-linked immunosorbent assay (ELISA) (F96 MaxiSorp, Thermo Fisher Scientific, Waltham, MA, USA) at a detection limit of 1 μg/kg and mean percentage recovery of 92.5% of AFB_1_ [47]. The method we utilized had been validated with naturally contaminated corn reference materials with the range 4.2 and 23.0 μg/kg AFB_1_ (product no. TR-A100, batch no A-C-268 and A-C 271; R-Biopharm AG, Darmstadt, Germany) and a recovery of 93% with relative standard deviation (RSD) of 4% and 2%, respectively. Briefly, the samples used in the validation were tested using a polyclonal antibody produced against AFB_1_-BSA. The secondary antibodies used were alkaline phosphatase-conjugated anti-rabbit antibodies (Sigma-Aldrich, St. Louis, MO, USA), and the substrate used was *para*-nitrophenyl phosphate (Sigma-Aldrich). The colorimetric reaction was measured using an ELISA plate reader (Multiskan reader, Thermo Fisher Scientific) using a 405-nm filter. Since confirmation of the presence of AFB_1_ in selected samples was crucial, the filtrate obtained underwent thin-layer chromatography using silica-gel-coated 20 × 20 cm glass plates (Fluka Analytical, Sigma-Aldrich), developed in chloroform:acetone (93:7, *v*/*v*) under vapor-saturated conditions, and detected directly under long-wave UV light based on fluorescence [48,49].

### 4.7. Dietary Exposure

Three parameters, probable daily intake (PDI), average probable daily intake (APDI) and maximum probable daily intake (MPDI), were used to assess dietary exposure level to aflatoxins. PDI and APDI estimates were calculated as described by Herrman and Yunes (1999) as follows [17]:PDI (ng/kg·bw/day) = maize intake (g/person/day) × levels of aflatoxins in the samples (μg/kg)/bw·(kg);APDI (ng/kg·bw/day) = maize intake (g/person/day) × average aflatoxin concentrations in the samples (μg/kg)/bw·(kg). The estimates of the maximum probable daily intake (MPDI) of aflatoxin were calculated using the formula:MPDI (ng/kg·bw/day) = (L × D)/bw (kg)where L is the 90th percentile concentration of aflatoxin in the samples, and D the daily consumption of maize-based foods (g/person/day).

For calculations of PDI, APDI, and MPDI, we utilized measured body weights. Various consumption levels were used to estimate exposure to aflatoxins. When the study by Abt Associates (2012) was considered, consumption levels were different based on regions and these values were used to estimate exposure for both maize and groundnut [50]. The average per capita consumption per day (based on FAO) and recommendations from the Tanzania Food and Nutrition Center (TFNC, 1997) were additionally used for maize [51].

### 4.8. Statistical Analyses

Statistical analyses were conducted using STATA 15.0 (Stata Corp., College Station, TX, USA). Categorical measures were expressed in numbers and corresponding percentages. European Union (EU) aflatoxin maximum limits for AFB_1_ contamination were 2 µg/kg and those of the EAC were 5 µg/kg [15] for all the collected grain and nut samples. 

Data collected on household assets were used to determine the household’s socioeconomic level. The wealth score was developed via principal component analysis (PCA) for continuous variables and multiple correspondence analysis (MCA) for categorical variables. The variables included in the development of this score were those that had a dominant modality with a frequency of less than 80% (household assets, such as radio, cart, television, bicycle; type of roofs and walls; presence of electricity; type of toilet and drinking water source). The coefficients of each linear combination were utilized as a weight for each variable. Distribution quartiles were used as the interval threshold to define the wealth index, with the 1st quartile representing the poorest and the 4th quartile representing the wealthiest section of the population. Total AFB_1_ contamination per household was obtained through the summation of the respective aflatoxin contamination per crop at household level. To investigate the risk of stunting and underweight due to aflatoxin exposure, logistic regression was used. The logistic regression models were utilized when investigating binary indicators such as stunting and underweight. Confounders such as age, gender, dietary diversity score, and wealth quintiles were also included in the model to obtain the odds ratios (OR) and 95% confidence intervals (CIs). Wasting was not included in the models, as only two children were wasted. For all the analyses, the *p* value cut-off level was set at <0.05.

## Figures and Tables

**Figure 1 toxins-15-00257-f001:**
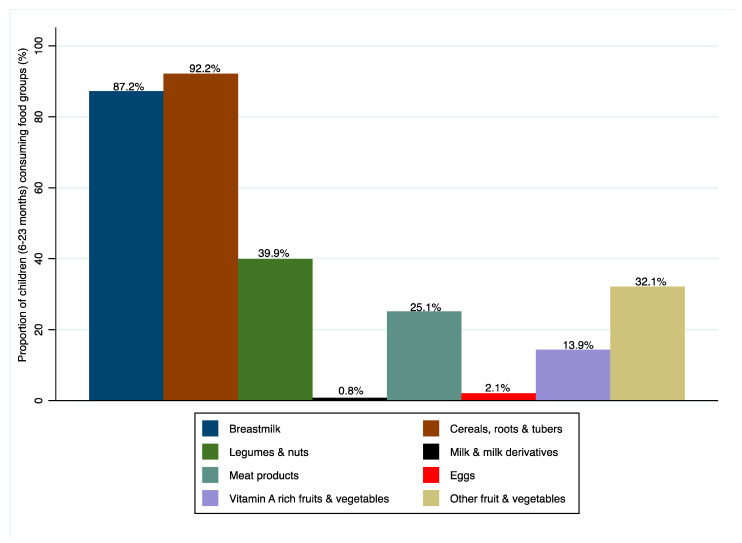
Proportion of children (6–23 months) consuming various food groups.

**Figure 2 toxins-15-00257-f002:**
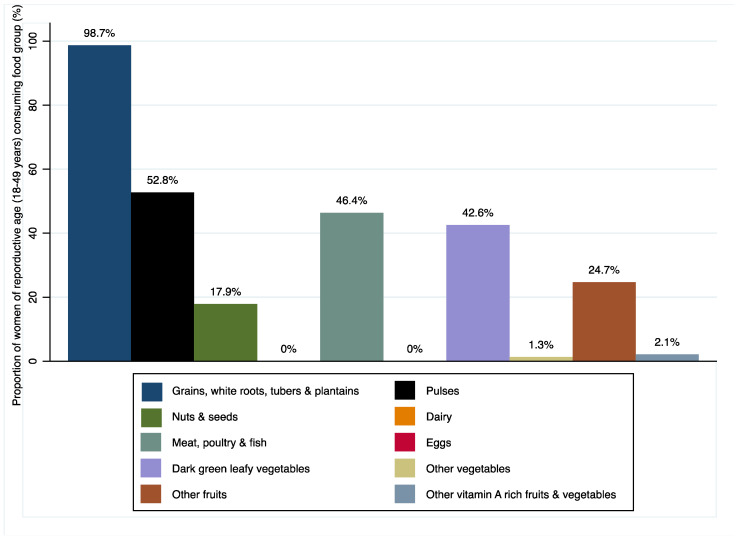
Proportion of women of reproductive age (18–49 years) consuming various food groups.

**Figure 3 toxins-15-00257-f003:**
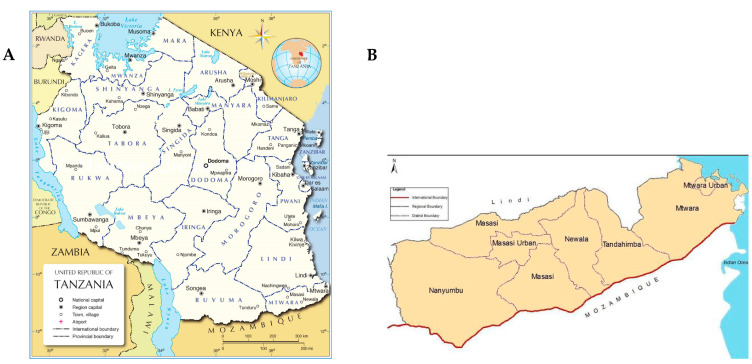
A map showing the location of (**A**) Mtwara in Tanzania and (**B**) the two study districts; Nanyumbu and Masasi, Mtwara South-eastern Tanzania.

**Table 1 toxins-15-00257-t001:** Characteristics of sampled mother–child pairs of Mtwara region, Tanzania.

Variable	Values	
** *Household characteristics* **		
Presence of grain store, n/N, %	98/243	40.3
Ownership of a radio, n/N, %	124/243	51.0
Ownership of a television, n/N, %	19/243	7.8
Ownership of phone, n/N, %	171/243	70.4
** *Child characteristics* **		
Age in months, median (range)	165	14.2 (6.0, 23.7)
Height for age Z-score, n, mean, (range)	164	−1.1 (−1.3, −0.9)
Weight for age Z-score, n, mean, (range)	164	−0.2 (−0.4, −0.1)
Weight for height Z-score, n, mean, (range)	164	0.5 (0.3, 0.6)
Stunting ^#^, n/N, %	34/164	20.7
Wasting ^††^, n/N, %	2/164	1.2
Underweight ^a^, n/N, %	7/164	4.3
** *Caregiver characteristics* **		
Age in years, median, (range)	241	27.7 (15, 45)
Weight in kg, n, mean (CI)	239	53.5 (52.3, 54.7)
BMI, in kg/m^2^, n, geometric mean (CI)	198	24.8 (20.8, 28.8)
Underweight ^a^, n/N, %	12/198	6.1
Normal weight ^b^, n/N, %	106/198	53.5
Overweight and obesity ^c^, n/N, %	80/198	40.4
** *Diet* **		
Proportion of children receiving complementary food before 6 months, n/N, %	8/165	5.0
Proportion of children receiving maize as part of complementary food, n/N, %	141/165	85.5
Proportion of children receiving groundnut as part of complementary food, n/N, %	13/165	7.8
Minimum dietary diversity—women, n, mean (CI)	234	3.1 (2.9, 3.2)
Met MDDW, n/N, %	75/234	32.2
Child dietary diversity score, n, mean (CI)	165	2.9 (2.8, 3.1)
Met minimum child dietary diversity score, n/N, (%)	8/165	4.8

^a^ Underweight was defined as body mass index less than 18.5 kg/m^2^ for caregivers and weight for age z-score of <−2 for children. ^b^ Normal weight for caregivers was defined as body mass index between 20–25 and for children weight for age z-score between −2 and 2. ^c^ Overweight and obesity was defined as body mass index between 25 and 30 for caregivers and for children weight for age z-score > 2. ^††^ Thinness was defined as BMI-for-age z-score of <−2. ^#^ Stunting was defined as height for age z-score < −2. n/N is used to calculate proportion of respondents associated with the specific characteristic where n indicates number of respondents that answer yes to the characteristics or meet MDDW/minimum child dietary diversity score and N is the total number of children or caregivers assessed for that characteristic.

**Table 2 toxins-15-00257-t002:** Aflatoxin B_1_ contamination in food samples from Mtwara region, Tanzania.

Sample	*n*	Median	*p* ^a^	Maximum	Samples Exceeding EU Regulatory Limit of 2 μg/kg: n/N (%)	Samples Exceeding EAC Regulatory Limit of 5 μg/kg: n/N (%)
Groundnut	87	2.8	-	1512	47/87 (54.0)	33/87 (37.9)
Bambara nut	24	1.6	0.44	148	9/24 (37.5)	7/24 (29.2)
Cassava	19	1.6	0.61	201	9/19 (47.4)	3/19 (15.8)
All maize	207	16.4	0.06	1507	165/217 (76.0)	140/217 (64.5)
*Maize Cobs*	5	4.1	0.68	32	2/5 (40)	3/5 (60.0)
*Maize Flour*	60	5	0.44	346	41/60 (68.3)	30/60 (50.0)
*Maize Grain*	152	11.3	0.003	2352	122/152 (80.3)	108/152 (71.1)
Sorghum	4	59.3	0.82	224	4/4 (100.0)	2/4 (50.0)

^a^ *p* values obtained from post hoc tests comparison of aflatoxin levels with aflatoxin in groundnut as the reference.

**Table 3 toxins-15-00257-t003:** Mean AFB_1_ concentration, probable daily intake (PDI), average probable daily intake (APDI) and maximum probable daily intake (MPDI) of aflatoxin B_1_ (AFB_1_) in adults of Mtwara region, Tanzania.

Food Item	Number (*n*) of Samples	Mean AFB_1_ (μg/kg) ± Standard Error	90th Percentile (μg/kg)	Consumption (g food day^−1^)	PDI Range ng/kg·bw/day ^d^	APDI ng/kg·bw/day ^e^	MPDI ng/kg·bw/day (90th Percentile) ^f^
Maize ^a^	212	17.2	391.1	429.4 ^b^	0–14, 571	1133	3906
Maize ^b^	212	17.2	391.1	771.0 ^a^	0–26, 162	2034	7699
Groundnut ^c^	78	4.1	48.5	15.1	0–375	14	18

^a^ Consumption rate recommended by Abt Associates = 429.4 g/person/day; ^b^ Per capita consumption per day recommended by TFNC = 771 g/person/day. ^c^ Groundnut consumption based on the report by Abt (2014); ^d^ probable daily intake (PDI), ^e^ average probable daily intake (APDI) and ^f^ maximum probable daily intake (MPDI).

**Table 4 toxins-15-00257-t004:** Predictors of stunting among children less than 24 months old in Mtwara region, Tanzania.

Outcome Measure	Stunting (*n* = 148)		Stunting (*n* = 148)		Height for Age Z-Score (*n* = 148)		Height for Age Z-Score (*n* = 148)	
	*Univariate*		*Multivariate*		*Univariate*		*Multivariate*	
Independent Variable	Odds Ratio (Confidence Interval)	*p*	Odds Ratio (Confidence Interval)	*p*	Beta (Confidence Interval)	*p*	Beta (Confidence Interval)	*p*
Age	1.14 (1.05, 1.24)	0.002	1.17 (1.06, 1.29)	0.001	−0.10 (−0.15, −0.06)	<0.0001	−0.12 (−0.16, −0.07)	0.00
Sex	1.18 (0.55, 2.55)	0.67	1.08 (0.46, 2.58)	0.85	−0.31 (−0.76, 0.14)	0.18	−0.41 (−0.86, 0.04)	0.08
Child Dietary Diversity	0.82 (0.53, 1.27)	0.34	0.94 (0.58, 1.52)	0.81	0.15 (−0.10, 0.39)	0.25	0.12 (−0.13, 0.37)	0.34
Wealth score	1.18 (0.92, 1.51)	0.62	1.19 (0.91, 1.55)	0.20	−0.05 (−0.18, 0.08)	0.43	−0.04 (−0.17, 0.08)	0.48
AFB_1_ concentration	1.00 (0.99, 1.00)	0.05	0.99 (0.99, 1.00)	0.05	0.00 (0.00003, 0.002)	0.04	0.001 (0.0003, 0.002)	0.04

Odds ratios, confidence intervals and *p* values were obtained via logistic regression models with the binary indicator stunting while beta values and confidence intervals were obtained via linear regression for the continuous outcome height for age z score. In the univariate model, the predictors age, gender, dietary diversity score, and wealth score were tested as independent predictor variables and included as confounders in the multivariate model.

**Table 5 toxins-15-00257-t005:** Predictors of wasting and weight for height age score among children less than 24 months in Mtwara region, Tanzania.

Outcome Measure	Wasting (*n* = 148)		Wasting (*n* = 148)		Weight for Height Z-Score (*n* = 148)		Weight for Height Z-Score (*n* = 148)	
	*Univariate*	*Multivariate*		*Univariate*		*Multivariate*		
Independent Variable	Odds Ratio (Confidence Interval)	*p*	Odds Ratio (Confidence Interval)	*p*	Beta (Confidence Interval)	*p*	Beta (Confidence Interval)	*p*
Age	0.48 (0.17, 1.36)	0.17	0.32 (0.05, 1.95)	0.22	−0.02 (−0.05, 0.02)	0.36	−0.01 (−0.05, 0.02)	0.47
Sex	1.08 (0.07, 17.5)	0.96	1.10 (0.03, 41.7)	0.96	0.01 (−0.36, 0.38)	0.96	0.01 (−0.38, 0.39)	0.97
Child Dietary Diversity	0.42 (0.07, 2.66)	0.36	0.22 (0.01, 5.32)	0.35	−0.03 (−0.23, 0.18)	0.79	−0.04 (−0.25, 0.17)	0.71
Wealth score	1.18 (0.47, 3.01)	0.72	1.14 (0.28, 4.59)	0.85	−0.06 (−0.16, 0.05)	0.29	−0.04 (−0.14, 0.07)	0.47
AFB_1_ concentration	1.00 (1.00, 1.004)	0.56	1.00 (1.00, 1.01)	0.27	−0.001 (−0.001, −0.0001)	0.02	0.001 (−0.001, −0.0001)	0.02

Odds ratios, confidence intervals, and *p* values were obtained via logistic regression models with the binary indicator wasting while beta values and confidence intervals were obtained via linear regression for the continuous outcome weight for height z score. In the univariate model predictors age, gender, dietary diversity score, and wealth score were tested as independent predictor variables and included as confounders in the multivariate model.

**Table 6 toxins-15-00257-t006:** Predictors of wasting among children less than 24 months old in Mtwara region, Tanzania.

Outcome Measure	Underweight (*n* = 148)		Underweight (*n* = 148)		Weight for Age Z-Score (*n* = 148)		Weight for Age Z-Score (*n* = 148)	
	*Univariate*		*Multivariate*		*Univariate*		*Multivariate*	
Independent Variable	Odds Ratio (Confidence Interval)	*p*	Odds Ratio (Confidence Interval)	*p*	Beta (Confidence Interval)	*p*	Beta (Confidence Interval)	*p*
Age	1.05 (0.90, 1.22)	0.56	1.05 (0.89, 1.23)	0.58	−0.06 (−0.09, −0.03)	0.001	−0.06 (−0.10, −0.03)	0.001
Sex	6.9 (0.81, 58.8)	0.08	8.1 (0.92, 71.4)	0.06	−0.16 (−0.50, 0.19)	0.37	−0.20 (−0.55, 0.15)	0.26
Child Dietary Diversity	0.73 (0.30, 1.78)	0.45	0.66 (0.25, 1.77)	0.41	0.06 (−0.13, 0.25)	0.54	0.03 (−0.16, 0.22)	0.76
Wealth score	1.12 (0.69, 1.82)	0.63	1.18 (0.71, 1.97)	0.53	−0.07 (−0.17, 0.02)	0.15	−0.05 (−0.15, 0.04)	0.27
AFB_1_ concentration	1.00 (0.99, 1.00)	0.38	0.99 (0.98, 1.01)	0.36	−0.0001 (−0.001, 0.0005)	0.69	−0.0001 (−0.001, 0.0004)	0.63

Odds ratios, confidence intervals, and *p* values were obtained via logistic regression models with the binary indicator underweight while beta values and confidence intervals were obtained via linear regression for the continuous outcome weight for height z score. In the univariate model predictors age, gender, dietary diversity score, and wealth score were tested as independent predictor variables and included as confounders in the multivariate model.

## Data Availability

The data presented in this study are available on request from the corresponding author. The data are not available due to institutional policies.

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
