# Peer review of "Aflatoxin Contamination, Exposure among Rural Smallholder Farming Tanzanian Mothers and Associations with Growth among Their Children"

_toxins, 2023, doi:10.3390/toxins15040257_

Round 1

Reviewer 1 Report

The work of Aflatoxin exposure among Tanzanian mother-child dyads from rural small holder farming households and associations with growth is very interesting. This research indicate the seriousness of aflatoxin contamination in foods consumed in the vulnerable population. The manuscript is based on a good experimental design and well-written. It requires only minor correction before publication in Toxins.

Please change AFB1 to AFB1.

Please change P<0.05 to P<0.05

P84, It would be better to provide more information about the two districts, such as residents' income and eating habits.

P154, The quantification limit and linear range of ELISA method for determination of aflatoxin B1 should be given.

P358-P361, Please add references

Author Response

Comments are greatly appreciated. Please find a point by point response to reviewer one comments.

Reviewer 2 Report

Dear Authors,

 I am pleased to have opportunity to review your manuscript describing the study on aflatoxin exposure among Tanzanian mother-child dyads from rural small holder farming households and associations with growth. The manuscript is interesting, well structured and can be read smoothly. However, some aspects shall be improved. The manuscript is recommended after corrections according to the comments bellow.

 1.       The title: I could not find the data on/assessment of exposure of mother-child dyads. In this case I suggest excluding the term “mother-child dyads” from the title.

2.       Line 14: I suggest giving the full names of EU and EAC before the abbreviations.

3.       Lines 18–19: I suggest changing “EU and EAC limits” to “EU and EAC limits, respectively”.

4.       Line 19: The wording “these previous observations” is not clear. Maybe, some text from lines 345–346 is missing?

5.       Lines 42–43: Please consider replacing “to consider” with “to be considered”.

6.       Lines 51–56: The text is not clear. Can it be made clearer by text from lines 404–406?

7.       Line 57: The reference (13) is not appropriate. The maximum limit is given in (31). However, (13) is no longer in force. It was repealed by Commission Implementing Regulation (EU) No 884/2014 and this one was repealed by Commission Implementing Regulation (EU) 2019/1793.

8.       Lines 63 and 337: I suggest changing “staple” to “staple food”.

9.       Lines 79–80: I suggest changing “less than 24 months” to “less than 24 months old”.

10.    Figure 1: I suggest making the arrow indicating Mtwara region in the left map more visible if possible.

11.    Lines 100–101 and 102: I see the texts “whether women 15-49” (100–101) and “respondents less than 15 years or greater than 49 years” contradictory. Please change if necessary.

12.    Line 115: Please change the wording “24 hour recall open recall”.

13.    Line 143: I do not find the sentence appropriate/complete. Please change it.

14.    Line 156: Please make it clear what “samples from Dedza” were.

15.    Line 161: It is mentioned that the extract was transferred to a conical flask. As I can conclude, the whole content from the blender (liquid and the solid sample) was transferred. If so, please correct (e.g., the mixture was transferred …). 

16.    Line 163: Please correct “for for AFB1”.

17.    Lines 184–191:  Please think over to synchronise the units in the equations.

18.    Line 243: Probably the wording “mass index less than for caregivers” is not complete.

19.    Line 245: I suggest changing “<-2 and <2” to “between – 2 and 2.

20.    Figure 1: It is difficult to distinguish which bar correspond to cereals, to milk and to eggs, because the colours (red, orange(?)) are very similar. I suggest changing two of them. The same in Figure 2 – it is difficult to distinguish which bar corresponds to pulses and which one to eggs.

21.    Lines 257 and 258: The full names are not necessary. The abbreviations EU and EAC suffice.

22.    Table 2: I suggest changing the title of the sixth and the seventh column, e.g., omitting the word “Percent” (e.g., Samples exceeding EU (EAC) regulatory limits of ….), because there is also n/N given in these two columns.

23.    Section 3.3: In Introduction section (lines 70–71) it is mentioned that the study would focus on the exposure among two groups (women of reproductive age and infants and young children). However, no description/data on AFB1 exposure of women and children specifically is given. Please improve.

24.    Tables 2 and 3, lines 270–280: I suggest rounding the numbers (e.g., 26161.7) to less significant figures.

25.    Table 3: PDI, APDI, MPD are given for adults. As mentioned above (section 3.3), the data for women and children are expected/should be given according to the statement in Introduction.

26.    Tables 4–6: Please explain what n=148 means (it is not the number of children included in the study? In the text, I can find that 250 households were selected and that each household had children less than 24 months old). Further, I suggest explaining the term odds ratio (in 3.3) and commenting it in 4. Discussion. Finally, is the wording “Total AFB1” appropriate? Maybe, “AFB1” or “AFB1 concentration” would be appropriate?

27.    Lines 464–465: I suggest removing the reference.

Author Response

The comments are greatly appreciated. Please find a point by point response of comments attached herein

Round 2

Reviewer 2 Report

Dear Authors,

The suggested changes were introduced following the suggestions or appropriate explanation was given.  However, the following minor changes are still suggested (I apologise if I overlooked them in the previous revision):

1.  Lines 15 and 17–18: The suggested full names of EU and EAC were given. However, they were placed in lines 17–18. I suggest moving them to line 15, where the abbreviations appear for the first time.

2.  Line 26: I think the word “indeed” in the sentence “indeed mycotoxin contamination in diets” is not appropriate. Please correct.

3.    Line 30: “Pakistan” should be “Tanzania”.

4. Lines 58, 218 and 219: The full names are not necessary. The abbreviations EU and EAC suffice.

5.   Table 2: probably “   Age, months (range)” should be “Age, n, months, (range)”?

6.    Lines 261– 266: I suggest giving explanation/note n/N means.    
